# When and Why Does Bias Mitigation Work?

**Abhilasha Ravichander**[1*]     **Joe Stacey**[2*]     **Marek Rei**[2]
[1]Allen Institute for Artificial Intelligence     [2]Imperial College London
abhilashar@allenai.org
{j.stacey20,marek.rei}@imperial.ac.uk

## Abstract

Neural models have been shown to exploit shallow surface features to perform language understanding tasks, rather than learning the deeper language understanding and reasoning skills that practitioners desire. Previous work has developed debiasing techniques to pressure models away from 'spurious' features or artifacts in datasets, with the goal of having models instead learn useful, task-relevant representations. However, what do models actually learn as a result of such debiasing procedures? In this work, we evaluate three model debiasing strategies, and through a set of carefully designed tests we show how debiasing can actually increase the model's reliance on hidden biases, instead of learning robust features that help it solve a task. Furthermore, we demonstrate how even debiasing models against all shallow features in a dataset may still not help models address a task. As a result, we suggest that only debiasing existing models may not be sufficient for many language understanding tasks, and future work should consider new learning paradigms to address complex challenges such as commonsense reasoning.

## 1 Introduction

Large-scale language models have established state-of-the-art performance on several language understanding tasks (Devlin et al., 2019; Liu et al., 2019; He et al., 2020). However, the performance of these models can degrade when presented with examples drawn from a different distribution than their training conditions. Past work has shown that these approaches are prone to exploiting spurious 'shortcuts' in the training data, instead of learning the intended task (Naik et al., 2018; McCoy et al., 2019; Geirhos et al., 2020).

Towards the primary goal of training NLP systems that can understand natural language, the propensity of neural models to rely on misleading

shortcuts presents a crucial obstacle to overcome. A number of debiasing approaches were proposed to prevent models from learning these shortcuts, including product-of-experts (Mahabadi et al., 2020), example reweighting (Clark et al., 2019) and confidence regularization (Utama et al., 2020a). These approaches are based on the intuition that by forcing models to avoid particular dataset shortcuts, we can achieve robust models that learn task-relevant features instead.

In this work, we critically examine this assumption. We take a step back and characterize the empirical behavior of debiasing methods at large. While prior work has largely demonstrated the effectiveness of debiasing methods by showing increased "robustness" on external challenge sets, where these improvements come from, the extent to which the model is unbiased, and the exact differences in behavior induced by debiasing procedures remain largely unknown.

Several factors make analyzing model behavior along these axes difficult in empirical settings. It is challenging to: (1) enumerate all possible biases in a dataset, (2) characterize task difficulty, and (3) develop fine-grained insights into the mechanisms models use to perform tasks— before and after debiasing procedures. We shed light on these questions, by constructing a suite of controlled tests to characterize the behavior of debiased models. We make our suite of synthetic tasks publicly available, providing a testbed for bias mitigation techniques.

Through the behaviors of debiased models on our controlled tasks, we identify several undesirable behaviors induced by debiasing procedures. First, we identify that debiasing can cause models to switch from the identified and targeted dataset bias to an unidentified dataset bias, while still seeming to demonstrate increased "robustness" to the targeted bias. This suggests that it is entirely possible that current bias mitigation approaches cause models to shift from relying on measurable biases

---
*The first two authors contributed equally to this work.

to relying on other unknown and unidentified biases in datasets (§5.1). Further, we characterize the complexity of a dataset[1] and demonstrate that even when all dataset biases have been addressed by a bias mitigation procedure, the model may not be able to learn the task if it is sufficiently complex (§5.2). This suggests that debiasing approaches alone are unlikely to help address complex tasks such as those involving commonsense reasoning. Our contributions are:

1. We provide a suite of six controlled datasets to probe model behavior when debiasing against a known bias (§4.1).[2]

2. For a range of different debiasing techniques and tasks, we show how debiasing can shift model reliance to hidden biases in the dataset, rather than the ones practitioners are aware of (§5.1).

3. We show for more complex tasks, debiasing alone does not result in more robust models (§5.2).

4. We highlight the strengths and weaknesses of three popular debiasing techniques, showing the circumstances when each technique may be appropriate (§5.3).

## 2   Related Work and Background

**Shortcut Learning**   Deep learning models are now known to be susceptible to "shortcut-learning", where models learn decision rules that appear superficially successful on benchmarks, but generalize poorly outside the training distribution (Geirhos et al., 2020). For example, a model trained to predict images that contain cows may use features from the background, such as the presence of grass, to make its prediction rather than learning to recognize cows (Beery et al., 2018). Much recent work in NLP has identified that models tend to capitalize on idiosyncrasies of a particular dataset at the expense of learning an underlying task. For the natural language inference task (Cooper et al., 1996; Dagan et al., 2006, 2013; Bowman et al., 2015; Williams et al., 2018), past work has found that NLI models make entailment decisions based on cues such as the degree of lexical overlap between pairs of sentences or the presence of words such as 'not' in the input (Dasgupta et al., 2018; Naik

---

[1]We consider datasets to be denotations for a task.

[2]Code/data available at https://github.com/AbhilashaRavichander/bias_mitigation.

et al., 2018; McCoy et al., 2019). Further, on the MultiNLI dataset (Williams et al., 2018), models can considerably outperform a random baseline by only looking at partial inputs (Gururangan et al., 2018; Poliak et al., 2018; Tsuchiya, 2018). Similar model behavior has been identified in several tasks which require natural language understanding, including argumentation mining (Niven and Kao, 2019), reading comprehension (Jia and Liang, 2017; Kaushik and Lipton, 2018), visual question answering (Zhang et al., 2016; Kafle and Kanan, 2017; Goyal et al., 2017; Agrawal et al., 2018), fact verification (Schuster et al., 2019), and story cloze completion (Schwartz et al., 2017; Cai et al., 2017).

**Bias Mitigation Methods**   Recent work has focused on 'debiasing' approaches, with the aim of learning unbiased models by pressuring them away from picking up dataset biases. These approaches can be broadly categorized as follows: (1) Data-centric approaches: procedures which manipulate the input data before running a standard model training procedure. By either filtering out biased instances, or by augmenting the dataset with additional examples, the model is expected to reduce its reliance on spurious biases in the dataset (Kaushik et al., 2020; Le Bras et al., 2020a; Yanaka et al., 2019; Min et al., 2020; Liu et al., 2020b; Wen et al., 2022). (2) Model-centric approaches: debiasing procedures that either modify the architecture of the model, the optimization, or the training procedure in order to make a model reduce its reliance on spurious biases (Sagawa et al., 2019; Tu et al., 2020; Mahabadi et al., 2021; Zhou and Bansal, 2020; Kirichenko et al., 2022; Wang et al., 2022; Du et al., 2022; Rajič et al., 2022; Wang et al., 2023; Ghaddar et al., 2021).

In this work, we will focus on model-centric debiasing procedures that can be specialized to particular tasks and datasets. Many popular model-centric approaches involve using the predictions of another model to influence learning. For example, several model-centric approaches use the predictions or the representations learned by a "shallow" classifier— a classifier that is lower capacity or impoverished in some other way such as accessing partial input—to construct a more robust model through debiasing (He et al., 2019; Clark et al., 2019; Mahabadi et al., 2020; Utama et al., 2020a,b; Sanh et al., 2021; Zhou and Bansal, 2020; Liu et al., 2020a; Clark et al., 2020; Xiong et al., 2021). The intuition behind these approaches is

that a "shallow" model is likely to rely more on dataset-specific biases. Popular approaches here include adversarial training (Belinkov et al., 2019a,b; Stacey et al., 2020), product-of-experts models where a 'main' model is trained in an ensemble with a shallow model (Clark et al., 2019, 2020; Utama et al., 2020b; Sanh et al., 2021), importance reweighting approaches where models are trained on a reweighted version of a source dataset (He et al., 2019; Utama et al., 2020b; Liu et al., 2020b; Mahabadi et al., 2020), or simply including a second round of finetuning on a more challenging subset of the data (Yaghoobzadeh et al., 2021). In this work, we consider whether debiasing strategies help (or not) with building better natural language understanding systems.

**Understanding the limitations of current debiasing methods** Amirkhani and Pilehvar (2021) show that using importance reweighting based on the prediction of a biased model may waste too much training data. Our work is similar, but examines fundamental questions about what models will learn from debiasing procedures. Mendelson and Belinkov (2021) show through a probing experiment that debiasing against a particular bias may increase the extent to which that bias is encoded in the inner representations of models. In this work, we study how debiasing procedures affect *model behavior*, as probe performance is not necessarily indicative of the information which a model actually uses to make predictive decisions (Ravichander et al., 2021; Elazar et al., 2021).

## 3 Framework and Definitions

### 3.1 Framework

Our goal is to analyze the mechanisms models employ during debiasing procedures. Our motivation is the discovery of shortcut-learning behavior in models for NLP tasks such as natural language inference and paraphrase detection. The question this work addresses is: when a model is debiased against one bias, what features does it learn?

In this work without loss of generality, we assume a binary paired-sequence classification task where a task-relevant feature $t$ is perfectly predictive of the label, and 'shortcut' features $w_1$ and $w_2$. Here, $w_1$ represents a superficially correlated feature that has either been identified, hypothesized to exist in a dataset, or inadvertently targeted by a broad-spectrum model, and $w_2$ is intended to repre-

sent an 'unknown' dataset bias. Our goal with these assumptions is to represent realistic conditions in NLP datasets, which contain possible shortcuts for models such as shallow lexical cues and annotation artifacts, but where the complete set of shortcuts a model can take is unknown. Our experiments focus on debiasing against $w_1$, representing the shortcuts that models are pressured away from taking by debiasing procedures (He et al., 2019; Clark et al., 2019; Utama et al., 2020b,a; Clark et al., 2020; Utama et al., 2020b; Sanh et al., 2021).

### 3.2 Definitions

Let $S$ be the space of all sequences (which are either English sentences or number sequences in our experimental setting). Let $y \in \{0, 1\}$ represent the classes in our binary paired-sequence classification task, intended to represent real-world paired sentence classification tasks such as NLI. The complete set of examples in the dataset $D$ is then drawn from the space $(x_1, x_2, y) \in S \times S \times \{0, 1\}$.

**Task features and spurious features** In our definition, a task-relevant feature $t$ is a feature that a dataset designer would identify as useful for performing a NLP task. [3] This motivates our choice to specify task-relevant features as those which correspond to a particular task definition. Conversely, a spurious feature $w$ is a feature that is considered not relevant for the task. In our work, we will discuss two kinds of spurious features— $w_1$ are spurious features that are targeted by debiasing methods i.e. *targeted bias*, and $w_2$ are spurious features that exist in the dataset but have not been debiased against i.e. *hidden bias*.

**Feature frequency and label correlation** We refer to frequency as the proportion of examples in a dataset containing a feature i.e $p(w)$. By predictiveness, we refer to the extent to which a feature correlates with a label i.e $p(y|w)$. For the spurious features $w_1$ and $w_2$, we set $w = 1$ as a spurious feature that is correlated with $y = 0$. To prevent the *absence* of this spurious feature from becoming an usable feature itself, we add an equal number

---

[3]We deviate from past definitions of task-relevant features such as that given in (Lovering et al., 2020). Lovering et al. (2020) consider task-relevant features as features where there exists some function $f$ such that $f(t(x)) = y$, i.e those features which are perfectly predictive of the task decision. However, such an empirical definition can be misleading due to both label noise in datasets, where a set of labels may not correspond to task-relevant features, as well as the possibility of multiple features that correspond to a particular task label.

| Task name | Task description | Task example |
|---|---|---|
| | Synthetic | |
| GREATER NUMBER | First number of seq. 1 > First number of seq. 2 | **Seq. 1:** 95789 68077 13030 37214 36462 56347 71940 65880 65242 35196 **Seq. 2:** 11453 67462 97337 49339 64738 30296 66671 69643 72790 4413 |
| SUM NUM-BERS | Sum of first 5 numbers in seq. 1 > sum of first 5 numbers in seq.2 | **Seq. 1:** 39440 66925 96563 64869 90299 90034 80263 98287 95837 76400 **Seq. 2:** 53169 59180 3453 79426 93135 92818 97662 35652 11135 81982 |
| FACTORIZATION | Sum of first 5 numbers in seq. 1 or seq. 2 is divisible by 7,11, or 13 | **Seq. 1:** 36620 96611 61508 79073 37465 21907 8158 7271 69979 39299 **Seq. 2:** 8631 11239 36336 74210 91520 33641 33775 26269 18699 81632 |
| | Naturalistic | |
| LEXICAL INFERENCE | Recognize synonym-antonym pairs | **Prem**: What the group hate most is the sickness., **Hyp**: What the group hate most is the illness. |
| LOGICAL INFERENCE | Reasoning with logical operators (AND/OR) | **Prem**: Stanley, Milo, Gabriel, Vinnie, Oakley and Eli are walking down the street, **Hyp**: Oakley or Alex, and Hudson or Elliot, and Stanley, and Eli, and Vinnie or Gabriel, and Milo are walking down the street. |
| COMMONSENSE INFERENCE | Pronoun resolution requiring commonsense inferences (Rahman and Ng, 2012) | **Prem**: In the story: The sniper shot the terrorist because he had orders, **Hyp**: In the story: The sniper shot the terrorist because the sniper had orders. |

Table 1: Task descriptions of six controlled datasets, both synthetic and naturalistic. All datasets are binary paired-sequence classification tasks. Salient attributes of the premise and hypothesis, or sequences, are highlighted.

of examples with the condition $w = 2$ as an instantiation of the same bias for $y = 1$. We refer to an example as *bias-neutral* if it contains neither of these properties i.e $w = 0$.

### 3.3 Characterizing Model Bias

If a model has learnt to use a particular feature, we expect that the model predictions will be consistent with the presence of that feature. By construction, we will design our datasets such that the spurious features are independent of each other, as well as the task-relevant feature. Then, we are interested in measuring the quantity $P(y'|w)$, where $y'$ is the *model prediction* and $w$ represents either a spurious or task-relevant feature. We can then write this as:

$$p(y'|w_1) = \sum_t \sum_{w_2} p(y'|w_1, w_2, t) \cdot p(w_2) \cdot p(t)$$

We expect this empirical probability to be close to 0.5 by assumption on our datasets, if a model is not influenced by a feature. We will say that a model is biased towards a particular feature, if this empirical probability $p(y|w)$ differs considerably from 0.5.

### 4 Experimental Setup

With the above definitions in place, our hypotheses are the following: (1) Debiasing procedures can cause pretrained language models to simply switch from relying on a measurable and targeted bias to relying on unknown, unmeasurable biases, (2) For higher-level complex tasks, debiasing a model does not itself help models learn tasks.

### 4.1 Data

It can be challenging to disentangle task-relevant features from all possible spurious artifacts, thus we construct datasets with tightly-controlled biases. The behavior of debiased models on each of these datasets is used to provide evidence for a particular hypothesis about a model's propensities, either in favor or against. We construct both synthetic datasets as well as naturalistic datasets. Our synthetic datasets consists of tasks where the input is a pair of number sequences, in order to better control for possible confounds from pretraining. Our naturalistic datasets are modeled after common NLP tasks such as commonsense reasoning, and lexical entailment.

**Synthetic Data** We construct simple synthetic binary paired-sequence classification tasks to study the debiasing procedures. Our tasks are intended to span varying levels of complexity. All our tasks feature pairs of k-length number sequences as input, and the output $y \in \{0, 1\}$. Our vocabulary $V$ consists of $\{0...\|V\|\}$. We set $k = 10$ and $V = 10000$.

Our three synthetic tasks are described in Table 1.

We introduce two kinds of biases: (1) token-level bias ($w_1$), where a particular token is correlated with a class, (2) overlap bias ($w_2$), where the co-occurrence of tokens across both input sequences is correlated with a class.

For $w_1$, in examples with the positive label class, p(y=1|$w_1$=1), the symbol '1' is added to one of the inputs. To balance this effect, we have a bias in the examples with the negative class, p(y=0, $w_1$=2), where the symbol '2' is added to one of the number sequences. The location of the bias and which sequence it is added to is randomly chosen, to any position except the last two numbers and the first number.

For $w_2$, in examples with the positive label class, p(y=1|$w_2$=1), the last number in both strings match. To balance this effect, we have a bias in the examples with the negative class, p(y=0, $w_2$=2), where the penultimate number in both number sequences match. If the example doesn't contain the bias, all numbers in both sequences will be unique.

**Naturalistic Data**  We create naturalistic datasets targeting lexical reasoning, logical reasoning or commonsense reasoning. Each dataset is created in the form of a sentence pair classification task. Examples are constructed to be highly lexically similar in order to mitigate the presence of additional dataset biases (besides the biases we introduce). Similar to the synthetic setting, we introduce two kinds of biases: (1) token-level bias ($w_1$), where a particular token or token sequence is correlated with a class, (2) overlap bias ($w_2$), where the co-occurrence of tokens across both input sequences is correlated with a class.

The lexical reasoning dataset targets synonym and antonym detection. We generate pairs using the template *'What the group hate most is the []'*. The entailment class is used as the label for synonym word-pairs and the non-entailment class is used as the label for antonym word-pairs. The $w_1$ trigger-word bias is introduced by replacing the word 'hate' with 'love' or 'like'.[4] The $w_2$ bias simulates a word-overlap bias. We append prefixes about belief to both premise and hypothesis, with prefixes matching for entailment and differing for non-entailment.[5]

---

[4]The word love correlates with the non-entailment class ($w_1$=2), while the word like correlates with the entailment class ($w_1$=1).

[5]'What we think' or 'What we believe'

The logical reasoning dataset evaluates model's abilities to correctly resolve logical expressions given truth values of individual variables. The premise lists the people doing a particular activity (truth values), while the hypothesis mentions some of the people doing the activity separated by AND/OR operators (logical expression). The $w_1$ bias is introduced by changing the activity mentioned to either 'running' (for entailment) or 'cycling' (for non-entailment). For the word overlap bias, a suffix is appended (specifying the activity is being performed 'once more'/'once again'), with only matching suffixes corresponding to the entailment class.

For commonsense reasoning we use the DPR dataset (Rahman and Ng, 2012; White et al., 2017), where the hypothesis contains a pronoun that resolves to one of two subjects in a premise. Resolving the pronoun correctly requires making commonsense inferences. $w_1$ and $w_2$ biases are introduced by inserting a prefix of *'In the story:'*. For the $w_1$ bias, the word *'in'* is replaced by *'within'* or *'inside'*. For the $w_2$ bias, the prefix is changed to *'In the story being told:'* or *'In the story being narrated:'*. The same prefix for both the hypothesis and premise corresponds to the entailment class.

## 4.2 Debiasing Procedures

The three debiasing approaches applied in this work are Product of Experts, Example Reweighting and Confidence Regularization (He et al., 2019; Clark et al., 2019; Utama et al., 2020b,a; Clark et al., 2020; Utama et al., 2020b; Sanh et al., 2021). Following previous work (Utama et al., 2020b; Ghaddar et al., 2021; Wang et al., 2023), we use BERT (Devlin et al., 2019) as the base model for debiasing. We assume teacher probabilities i.e probabilities assigned by a model that relies on $w_1$ to make predictions. When the biased feature does not occur ($w_1 = 0$), this probability is set to 0.5.

**Importance Reweighting**  This method weights training examples based on the presence of biased features within each example, down-weighting examples with stronger known biases. The loss for each minibatch B is:

$$\mathcal{L}_B = \sum_{i=1}^{b} CELoss(\hat{y}_i) \cdot \frac{1 - p_i}{\sum_{j=1}^{b} 1 - p_j}$$

where $p_i$ is the teacher probability that the $i$th observation is the correct class, $\hat{y}_i$ are the predicted

| Task | Bias 1- token correlation bias | | | | Bias 2- token overlap bias | | | | Unbiased Test Set | | | |
|---|---|---|---|---|---|---|---|---|---|---|---|---|
| | Baseline | Reweight | PoE | CR | Baseline | Reweight | PoE | CR | Baseline | Reweight | PoE | CR |
| Synthetic | | | | | | | | | | | | |
| Find Greater Number | 0.986 | 0.501 | 0.502 | 0.5245 | 0.534 | 0.590 | 0.603 | 0.5867 | **0.945** | 0.940 | 0.940 | 0.855 |
| Sum Numbers | 0.978 | 0.499 | 0.499 | 0.5904 | 0.606 | 0.904 | 0.627 | 0.5625 | **0.830** | 0.801 | 0.808 | 0.768 |
| Factorization | 0.979 | 0.501 | 0.502 | 0.5472 | 0.687 | 0.995 | 0.992 | 0.7604 | 0.499 | 0.499 | **0.500** | 0.499 |
| Natural | | | | | | | | | | | | |
| Lexical Inference | 0.902 | 0.486 | 0.495 | 0.418 | 0.750 | 0.999 | 1.000 | 0.825 | 0.676 | 0.671 | **0.701** | 0.666 |
| Logical Inference | 0.847 | 0.492 | 0.495 | 0.553 | 0.637 | 0.904 | 0.746 | 0.627 | **1.000** | 0.997 | 0.938 | 0.991 |
| Commonsense Inference | 0.924 | 0.474 | 0.477 | 0.683 | 0.727 | 1.000 | 1.000 | 0.652 | 0.502 | **0.511** | 0.506 | 0.498 |

Table 2: Bias probabilities show the extent to which the model prediction correlates with the value of a biased feature. Reweight, PoE and CR represent the three debiasing procedures in this study, and Baseline represents the model where no debiasing has been performed. Debiasing a model against one bias ($w_1$) may cause it to overrely on a different bias ($w_2$). The unbiased test set does not contain spurious features $w_1$ or $w_2$. All results are averaged over five seeds.

model probabilities for the $i$-th observation, and $b$ is the number of examples within the minibatch (Clark et al., 2019; Utama et al., 2020b).

**Product-of-Experts** The Product of Experts (PoE) approach combines the model probabilities with the teacher probabilities, incentivizing the model to learn from features other than the bias features. The loss for each example $i$ is:

$$\mathcal{L}_i = CELoss(\sigma(\log p_i + \log \hat{y}_i))$$

where $p_i$ are the teacher probabilities for both classes and $\hat{y}_i$ are the predicted model probabilities for $i$ (Clark et al., 2019; Utama et al., 2020b; Mahabadi et al., 2020). The loss $Loss_B$ for each minibatch is the average loss of each observation.

**Confidence Regularization** Confidence Regularization (CR) uses the bias teacher probabilities $p_i$, representing the teacher probability that the observation is the correct class, in addition to model predictions $\hat{z_{ik}}$ that observation $i$ is class $k$ before any debiasing has been applied. The confidence of the model predictions before the debiasing are regularized based on the teacher probabilities $p_i$, incentivizing the model to be less confident for more biased examples. A robust model is then trained through self-distillation using both $p_i$ and $\hat{z_{ik}}$, with the loss for an example $i$ is:

$$\mathcal{L}_i = -\sum_{k=0,1} \frac{\hat{z_{ik}}^{1-p_i}}{\hat{z_{ik}}^{1-p_i} + (1-\hat{z_{ik}})^{1-p_i}} \cdot \log \hat{y_{ik}}$$

The loss $\mathcal{L}_B$ for each minibatch is the average loss from each observation.

## 5 Results and Analysis

### 5.1 What does debiasing do?

We consider datasets for each task which contain both $w_1$ and $w_2$ as dataset biases. Recall that $w_1$ represents a known bias which is the target of the debiasing procedure, and $w_2$ represents an unidentified shortcut, and thus is not within the scope of the debiasing procedure. We find that debiasing the model against the $w_1$ bias consistently increases the model's reliance on the $w_2$ bias (Table 2).

The *extent* to which the model relies on this secondary bias depends on the difficulty of the task (Table 2). We observe that for both example-reweighting and PoE, the model relies almost completely on the $w_2$ bias for the most difficult tasks in each category (FACTORIZATION and COMMONSENSE INFERENCE). In comparison, the confidence regularization procedure incentivizes a model to be uncertain on biased predictions. and we observe that on the most challenging tasks— the model when debiased from $w_1$, picks up neither $w_2$, nor learns to perform the task. For the simpler tasks (FIND GREATER NUMBER, SUM NUMBERS, LEXICAL INFERENCE, LOGICAL INFERENCE), we find that the models rely on $w_2$ to a lesser extent. but only seldom gains improvements on the task.

A reasonable concern here is that these conclusions would not generalize to newer models, and that newer models would neither exploit the bias $w_1$, nor switch to relying on the hidden bias $w_2$. Due to the considerable computational cost involved of training several debiased models, we only experiment with DeBERTa-large (He et al., 2021). We find that our conclusions generalize when using either a BERT or DeBERTa model, these results can be found in the appendix (Fig. 2).

| | Task MDL | Bias 1- token correlation bias | | | | Unbiased Test Set | | | |
|---|---|---|---|---|---|---|---|---|---|
| Task | | Baseline | Reweight | PoE | CR | Baseline | Reweight | PoE | CR |
| *Synthetic* | | | | | | | | | |
| Find Greater Number | 3877.35 | 0.892 | 0.5024 | 0.5028 | 0.5008 | 86.73% | **95.51%** | 95.19% | 79.6% |
| Sum Numbers | 7179.58 | 0.9788 | 0.5025 | 0.5012 | 0.514 | **85.3%** | 83.15% | 69.84% | 79.03% |
| Factorization | 16426.14 | 0.9793 | 0.5009 | 0.5 | 0.5897 | 50.10% | 49.98% | **50.15%** | 49.94% |
| *Natural* | | | | | | | | | |
| Lexical Inference | 460.98 | 1.000 | 0.501 | 0.521 | 0.458 | **72.10%** | 68.40% | 70.67% | 65.80% |
| Logical Inference | 477.19 | 0.985 | 0.496 | 0.497 | 0.478 | **99.89%** | 90.02% | 97.58% | 99.61% |
| Commonsense Inference | 480.19 | 1.000 | 0.493 | 0.486 | 0.586 | 50.33% | 51.60% | **52.77%** | 49.99% |

Table 3: Debiasing a model against one bias ($w_1$) does not indicate a model will learn a task. Bias probabilities reflect the extent to which model predictions correlate with the biased feature. The unbiased test set does not contain $w_1$. Even when debiasing procedures are effective and model predictions are not highly correlated with the targeted feature ($w_1$), task performance does not always increase.

## 5.2 Debiasing in the limit

We ask if, hypothetically, a model were to be debiased against all the spurious correlations that existed in a dataset— would the model then be able to learn the task? To put it provocatively, our hypothesis is that without shortcuts there is no learning that the models we study can do for more complex inference tasks.

In order to evaluate this hypothesis, we construct synthetic datasets such that $w_1$— the known token correlation bias— is the only shortcut. We quantitatively measure the extent to which some tasks are easier to learn than others from input text. Inspired by Voita and Titov (2020), we quantify this complexity using the online minimum description length (MDL) (Rissanen, 1978; Voita and Titov, 2020; Lovering et al., 2020), in addition to accuracy. We find that implementing any debiasing method does not enable the model to learn complex tasks. This is the case even when the dataset does not contain additional $w_2$ bias (Table 3). For the commonsense and the factorization tasks, performance remains close to a majority baseline.

## 5.3 How do debiasing procedures differ?

Each debiasing method has different strengths and weaknesses, and there is little evidence to suggest which procedure would be most effective in a given data regime. We control three parameters of a dataset: the frequency of biases in the dataset, the dataset size and label noise. We study four debiasing procedures in these different data scenarios— example reweighting, PoE, confidence regularization, and dataset filtering (Wu et al., 2022; Le Bras et al., 2020b), a special case of example reweighting where the biased examples are completely re-

moved from the datasets.[6][7]

**Bias Frequency:** We measure the performance of each debiasing method when changing the frequency of the biases. We see the performance of dataset filtering degrades the most, as the frequency of the biased examples increases (Fig. 1 - a). As this technique filters biased examples, the more biased examples there are, the fewer observations are available for the model to learn from.

**Dataset size:** We compare debiasing procedures in high-resource and low-resource scenarios. The larger the dataset, the more effective PoE and Reweighting are (Fig. 1 - b). This suggests that as the number of examples increases, the ability of debiased models to ignore biased features improves. This is not the case for data filtering, where all biased examples are simply removed, and therefore the bias never influences the model predictions (Fig. 1 - b).

**Label Noise:** We consider the effect of label noise in a dataset. In many real-world datasets, a small percentage of instances can be expected to be ambiguous or wrongly labeled. As the amount of label noise increases, the baseline and CR increasingly rely on the $w_1$-bias (Fig. 1 - c Graph 1), whereas all other debiasing methods rely more on the $w_2$-bias (Fig. 1 - c Graph 2). Data filtering relies more on the $w_2$-bias than any other method, as this method is unable to learn from the $w_1$-bias which is entirely filtered out (Fig. 1 - c Graph 2).

---

[6]This is equivalent to zeroing out the gradients for these examples under the example reweighting paradigm.

[7]To explore the differences between example reweighting and dataset filtering, we set $w_1$ and $w_2$ to be 90% correlated with the class label since we use a counts-based bias model.

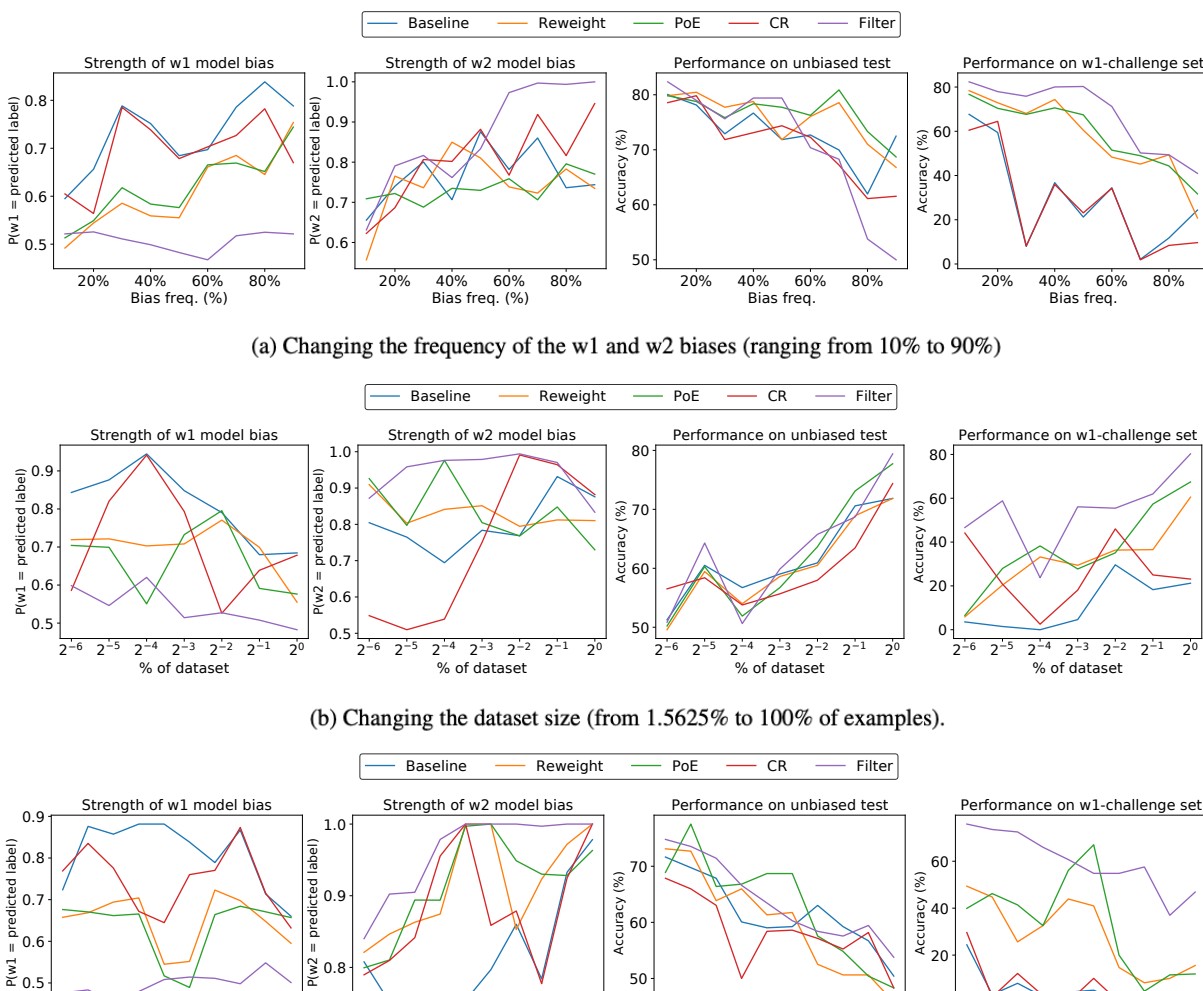

(a) Changing the frequency of the w1 and w2 biases (ranging from 10% to 90%)

(b) Changing the dataset size (from 1.5625% to 100% of examples).

(c) Changing the amount of label noise from 5% to 50% of examples.

Figure 1: Debiasing performance for lexical entailment, stratified by bias frequency, dataset size, and label noise.

## 6 Discussion

We briefly discuss our findings and recommendations for debiasing methods going forward.

**Which debiasing technique to choose?** In our work, we shed light on how different debiasing techniques can be effective based on properties of the training dataset, including how frequent biases are, the size of the dataset as well as the amount of label noise in the data. We find that confidence regularization performs worse than the other debiasing methods in a variety of data regimes we evaluate, sometimes performing little better than the baseline. PoE and reweighting often exhibit similar behavior. Data filtering performs worst in highly-biased datasets.

In addition, practitioners may want to consult several metrics while making the decision of which

debiasing technique to choose, such as: (1) Performance on out-of-domain challenge sets, (2) Performance on long-tailed or rare phenomena, (3) Number of additional parameters introduced by the debiasing procedure, (4) Additional time required during training/inference, (5) Efficiency of the debiasing procedure (Schwartz et al., 2020).

**Invest in robust evaluation** This work demonstrates that debiasing can cause models to switch from relying on a known and measurable bias, to overrelying on hidden biases in datasets. This suggests that practitioners would benefit from constructing a wider range of evaluations to diagnose model reliance on spurious features in datasets, as well as methods to automatically identify these biases.

**Why use debiasing?** If the underlying task is accessible to a model but the model exploits shortcuts, debiasing can be effective. However, if the underlying task is not accessible to models in the first place, debiasing procedures alone cannot be expected to be a legitimate path to making progress on the task.

**Computational Reasoning** Our work shows that for complex tasks, debiasing existing models may not be sufficient. This calls for investment in new learning paradigms that can help address complex tasks.

**Synthetic Datasets** Our work utilizes synthetic datasets, with controlled bias settings to enable studying the behavior of debiasing approaches. These datasets allow us to validate specific hypotheses about model behavior which are challenging to study with real-world datasets where there might be several bias features, or combinations of bias features, that a model can potentially exploit to address a task. Future work can consider designing synthetic datasets as a tool to control for confounding factors and evaluate hypotheses about model behavior.

## 7 Conclusion

NLU models have been shown to overrely on spurious features. This has lead to the development of several debiasing procedures to pressure models away from this behavior. In this work, we show that many of these procedures may simply push models to rely on a different set of shortcuts, and that the existing models may not be able to address complex NLU tasks even after all the spurious features have been addressed. We hope our work serves as a foundation to evaluate the effectiveness of debiasing procedures, sheds light on how these methods work, and provides an impetus for new classes of models for complex NLU tasks.

## 8 Limitations

We specify assumptions used in this study and highlight limitations of these assumptions.

1. We assume the task-specific label perfectly corresponds with a task label in all our dataset constructions. In the real world however, there is likely to be some amount of label-noise, particularly in large-scale datasets.

2. We perform our experiments on tightly controlled, largely synthetic datasets where we can control both the task-relevant features as well as possible shortcuts. We do this as real-world datasets could have several possible interacting confounds. Our settings provide a controlled environment that then makes it feasible to study the behavior of debiasing approaches which may not have been possible otherwise, at some expense of generalization.

3. Our datasets are constructed with a maximum of two biases. In practice, datasets may contain several biases which are usually unknown. The datasets used in this study are intended to represent a proof-of-concept of these scenarios to better understand model behavior.

4. We look at methods of debiasing against a specific bias. Some broad-spectrum debiasing methods do not target a particular shortcut, but rather target several possible shortcuts in the dataset that a particular 'shallow' model has access to. In our framework, shortcuts $w_1$ which are intended to refer to a specific bias, can also represent a class of biases a 'shallow' model has access to.

5. Pretraining may affect dataset conditions, by inducing unanticipated correlations between tokens. In order to control for this behavior, we construct six different datasets using both natural language, as well as numeric symbols.

6. We investigate supervised settings in this study. We leave studying biases in few-shot and zero-shot learning settings for future work.

## Acknowledgements

The authors would like to thank Aditya Potukuchi, Aakanksha Naik, Tejas Srinivasan, and Nishant Subramani for valuable discussions related to this project. The authors would also like to thank the three anonymous reviewers for their valuable feedback.

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

# A  Dataset Construction Procedures

We include expanded details of the construction procedures for the naturalistic datasets in our study.

**Lexical reasoning**  The lexical reasoning task involves reasoning whether a pair of words are either antonyms or synonyms. Examples are created in the form of NLI using the template: 'What the group hate most is the [word]', with this phrase repeated for both the hypothesis and the premise where the synonym or antonym words replace the [word] tag. We use the antonyms and synonyms of nouns available on WordNet.

The $w_1$ bias is inserted by changing the word 'hate' in the template for either 'like' or 'love'. When the word 'hate' is used, the example is neutral with respect to $w_1$, while the word 'like' positively correlates with the entailment class and the word 'love' positively correlates with the non-entailment class. The $w_2$ word-overlap bias is inserted by adding the text 'What we believe' or 'What we think' at the beginning of both sentences. For the $w_2$ bias, the entailment class positively correlates with examples where this additional text is the same for the premise and hypothesis. The non-entailment class on the other hand positively correlates with examples that have different text added at the beginning of the sentences. Examples that are neutral with respect to the $w_2$ word-overlap bias do not contain any additional text, in either the hypothesis or the premise.

**Logical reasoning**  The logical reasoning dataset contains a premise expressing a list of people who are undertaking an activity ('walking down a street'), while the hypothesis uses a combination of AND and OR logical operators to also express which people are doing this activity. The NLI class is determined by reasoning over these AND and

|                        | #Train          | #Dev          | #Test           |
|------------------------|-----------------|---------------|-----------------|
| GREATER NUMBER         | 5000 (50%)      | 1000 (50%)    | 16374 (50.16%)  |
| SUM NUMBERS            | 5000 (50%)      | 1000 (50%)    | 16533 (50.29%)  |
| FACTORIZATION          | 5000 (50%)      | 1000 (50%)    | 16417 (50.19%)  |
| LEXICAL INFERENCE      | 2,808 (50.00%)  | 472 (50.00%)  | 476 (50.00%)    |
| LOGICAL INFERENCE      | 20,000 (50.55%) | 5,000 (50.16%)| 5,000 (50.54%)  |
| COMMONSENSE INFERENCE  | 2,080 (50.00%)  | 486 (50.21%)  | 1095 (50.05%)   |

Table 4: The number of instances in each dataset across splits (Train/Dev/Test), as well as the proporition of instances of the majority class (in parentheses).

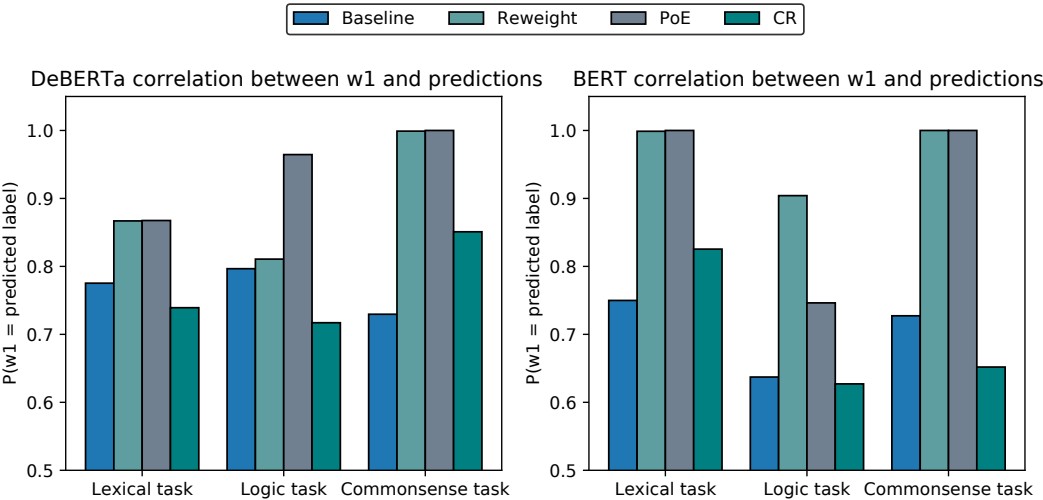

Figure 2: Comparison of BERT and DeBERTA behavior, in terms of the proportion of examples where the model prediction is the same as the direction of the $w_2$ word-overlap bias. 1.0 indicates that the $w_2$ bias fully determines the model predictions, compared to 0.5 where the $w_2$ bias is not correlated with the model predictions.

OR operators to determine the truth of the hypothesis based on the premise. We avoid using two OR statements consecutively to ensure that the hypotheses is unambiguous while also being written fluently in natural language.

Both the training and test data contain between 6 and 20 logical operators for each hypothesis. While the training data uses exactly 2 OR statements in each hypothesis, the test data contains three or more OR statements. The names of the people undertaking the activity are also different in the training and test data.

The $w_1$ word correlation bias is inserted into the dataset by changing the activity being performed, with 'running' correlating with the entailment class, and 'cycling' correlating with the non-entailment class. When the activity remains 'walking', then the observation is neutral with respect to $w_1$. The $w_2$ word overlap bias involves adding the words 'once more' or 'once again' to the end of the sen-

tences, with these phrases overlapping between the hypothesis and premise for entailment examples. Examples that are neutral with respect to $w_2$ do not include any additional text at the end of either sentence.

**Commonsense reasoning** The commonsense reasoning dataset involves inserting the $w_1$ and $w_2$ biases into the DPR dataset (White et al., 2017; Rahman and Ng, 2012).

The $w_1$ and $w_2$ biases are inserted by adding additional text at the beginning of each sentence without changing its meaning. We start each sentence with 'In the story', replacing the word 'In' with either 'Inside' (correlating with the entailment class) or with 'Within' (correlating with the non-entailment class). The word 'In' is unchanged for examples that are neutral to the $w_1$ bias. The $w_2$ bias is inserted by changing the start of each sentence to be either 'In the story being narrated' or 'In the story being told', with the entailment class pos-

itively correlating to examples where the phrase is the same across both sentences. Examples that are neutral to the $w_2$ bias do not include this additional text.

## B    Dataset Statistics and Analysis

We include detailed statistics of all six controlled synthetic datasets in Table. 4. For computing the MDL over naturalistic datasets, we consider 476 examples from each unbiased test set to ensure tasks are comparable.

## C    To what extent do our conclusions generalize to newer models?

While we follow prior work in studying the effect of debiasing procedures on BERT (Utama et al., 2020b; Ghaddar et al., 2021; Wang et al., 2023), we also wish to examine to what extent our conclusions generalize to newer models. We examine the performance of all three debiasing procedures on the three naturalistic datasets constructed in this study, finding that BERT and DeBERTA exhibit largely similar behavior under the effect of debiasing procedures.