# OpenReview forum: "When and Why Does Bias Mitigation Work?"
_EMNLP/2023/Conference — EMNLP 2023 Findings_

### Official Review · Reviewer_BTLE · 2023-07-22

**Soundness:** 4

**Excitement:**

4: Strong: This paper deepens the understanding of some phenomenon or lowers the barriers to an existing research direction.

**Paper Topic And Main Contributions:**

The paper uses three carefully-designed experiments to probe model debiasing strategies. In these experiments, the "hidden biases" can be understood in an interpretable form. The paper shows that debiasing increases the model's reliance on hidden biases instead of robust features, which corrects the direction of future work for debiasing models and addressing more complex tasks.

**Reasons To Accept:**

1. The synthetic experiment design is very elegant and useful for detecting the model's debiasing behavior;
2. Running thorough experiments to compare different debiasing methods in different dataset properties;

**Reasons To Reject:**

1. lack the mathematical analysis for each debiasing method's performance;

**Reproducibility:**

4: Could mostly reproduce the results, but there may be some variation because of sample variance or minor variations in their interpretation of the protocol or method.

**Reviewer Confidence:**

4: Quite sure. I tried to check the important points carefully. It's unlikely, though conceivable, that I missed something that should affect my ratings.

---

> ### Author Rebuttal · Authors · 2023-08-29
>
> **Thank you for engaging with our work, and championing our paper!**
>
> **We are pleased you noted our carefully-designed experiments and found that this work will correct the direction of future work on debiasing.**
>
>  **Thank you particularly for noting that our experimental design is elegant and thorough.**
>
>
> > Lack of mathematical analysis for debiasing method’s performance
>
> We contribute a mathematical framework both for measuring model bias (L 274-289), and for measuring task difficulty (L 458- 465). If there is any additional mathematical analysis you would like us to include please let us know, and we would be happy to address this in the final version.

---

### Official Review · Reviewer_FLhX · 2023-08-04

**Soundness:** 3

**Excitement:**

3: Ambivalent: It has merits (e.g., it reports state-of-the-art results, the idea is nice), but there are key weaknesses (e.g., it describes incremental work), and it can significantly benefit from another round of revision. However, I won't object to accepting it if my co-reviewers champion it.

**Paper Topic And Main Contributions:**

This paper examines three representative debiasing techniques to understand what models actually learn as a result of debiasing. They found current bias mitigation approaches cause models to shift from relying on measurable biases to relying on other unknown and unidentified biases in datasets.

**Questions For The Authors:**

Question A: Why do you expect debiasing would help address a task (Line 19)? The purpose of debiasing seems to just remove toxic biases from the model when deployed to the general public.

Question B: Why not utilize existing debiasing datasets? What is the motivation and novelty of creating naturalistic data by themselves? For example, one possible issue is that using template “What the group hate most is the <>” might make the debiased model over-adapt to this sentence and shift model’s distribution away from a more realistic data distribution (of NLI); and the debiasing techniques were targeting at those more realistic data distributions. Similar things hold for other naturalistic dataset.

Question C: Synthetic dataset seems a bit weird, because it’s very different from the distribution a LLM is used to.

Question D: the instance will be ungrammatical if you remove w_2 in the data and make the task unresolveable by nature. How do the authors deal with this? For example, in Table 1 – “Lexical Inference”, the sample would be:
**Prem**: What the group hate most is the sickness., **Hyp**: What the group hate most is the [“illness” deleted].

**Reasons To Accept:**

The paper is well written and the message is clear.

The motivation of the paper could shed light on generative models.

Interesting observation of the model keeps looking for spurious correlation to finish the task.

**Reasons To Reject:**

Methodology and experiment design have some key problems and thus lack a strong support for the conclusion.

The setting seems toy — encoder, NLI; and has limited implication for more powerful generative models.

See questions for the author below.

**Reproducibility:**

5: Could easily reproduce the results.

**Reviewer Confidence:**

3: Pretty sure, but there's a chance I missed something. Although I have a good feel for this area in general, I did not carefully check the paper's details, e.g., the math, experimental design, or novelty.

**Typos Grammar Style And Presentation Improvements:**

Line 322, 324 330, 332: math equation rendering

Figure 1 has too many details and y axis (and the subtitle) means different things (in a single row); this makes the plot hard to interpret.

In section 5.3, the author starts off by trying to figure out which debiasing procedure would be most effective, but then this section doesn’t give an answer. Instead, the beginning of section 6 gives the takeaway.

---

> ### Author Rebuttal · Authors · 2023-08-29
>
> **Thank you for your review and for engaging with this work! We are glad you found our paper to be well-written and our empirical findings interesting.**
>
> > Why do you expect debiasing would help address a task? The purpose of debiasing seems to just remove toxic biases from the model when deployed to the general public.
>
> The goal of debiasing is to prevent models from relying on different types of spurious biases, thereby guiding the model to focus only on features actually relevant to the task and ideally also lead to better performance [L47-50]. Refer to our literature review L143-L164.  Toxic biases are only a subset of spurious biases.
>
> > Why not utilize existing debiasing datasets?
>
> We do not use existing debiasing datasets, as we want to carefully and systematically control several factors in order to study the behavior of debiasing methods.
>
> Some of these considerations are:
> 1. **Number of bias features that correlate with the task feature:** Our research question requires two, controlled dataset biases. One is a dataset bias that the model is initially de-biased against, and a second latent bias. This is not a setting investigated in existing debiasing datasets,
> 2. **Avoiding confounding dataset biases:** NLP datasets contain several possible features that are challenging to disentangle. As stated in Zhou and Bansal 2020,  “In reality, almost every dataset contains countless such diverse [dataset] biases.”.
> 3. **Constructing examples that are neutral with respect to each dataset bias:** To assess the performance of a model on the underlying task, we need to have the ability to test the model on examples that do not contain any dataset biases.
> 4. **Strength of correlation between the dataset biases and each class, and frequency of the dataset biases:** Adjusting the strength and frequency of the biases allows us to understand the conditions under which each different de-biasing method is effective. This is not possible when using or modifying an existing de-biasing dataset.
> 5. **Comparing de-biasing methods on a range of different tasks (with a range of difficulties):** By constructing synthetic simulation data, we are able to construct datasets for tasks free of confounding bias features, which in turn allows us to measure the difficulty of the underlying task itself for models.
>
> > Synthetic datasets seem weird
>
> 1. Constructing synthetic datasets is a well-established tool in the literature for probing and studying model behavior. For example, Naik et al. 2019 used a series of templates to generate challenge datasets that surfaced shortcut-learning behavior in NLI models.  SImilarly McCoy et al. 2020 contribute a ‘controlled evaluation set’ called HANS to identify model reliance on heuristics. Another notable example is that of Lovering et al. 2020, who use both symbolic and naturalistic data to study the inductive biases captured by pretrained language models. Yet another example is Sinha et al. 2021, who studied model behavior on permuted sentences to examine how models process syntax.
> 2. Our synthetic simulation tasks are constructed to be close to natural data. Several of the tasks we use in our simulation  include features relevant to real NLP tasks (DPR commonsense reasoning [Rahman and Ng, 2012], Lexical Entailment).  The bias feature types we use have been substantiated in the literature (word overlap and tigger word biases [Dasgupta et al., 2018; Naik 129 et al., 2018; McCoy et al., 2019.]) (L 128-129; L 348-351)
> 3. The synthetic simulation we build allows us to control several factors that have thus far made it challenging to conduct a similar analysis of debiasing methods. In most NLP datasets, there may be several interacting confounds  [L 61-71], and it is challenging to estimate the number or type of bias features, the correlation of these features with the task, the frequency of the bias features in the dataset, and to estimate the difficulty of the underlying task itself. Our simulation allows us to systematically control all these factors, which allowed us to study the actual behavior of debiasing methods in a way that had not been done before.
>
> > Encoder and NLI Settings
>
> We study existing debiasing methods. We do not propose new debiasing methods in this study. We thus construct our simulations to imitate settings where debiasing approaches thus far have been shown to be effective (sentence -pair classification tasks with encoder-only models such as BERT and RoBERTa as the primary model, see Mendelson & Belinkov, 2021, Wang et al., 2023, He et al., 2019; Clark et al., 2019; Mahabadi et al., 2020; Utama et al., 2020a,b; Sanh et al., 2021; Zhou and Bansal, 2020; Liu et al., 2020a; Clark et al., 2020;).
>
> > Naturalistic instances will be ungrammatical when removing w_2
>
> **Naturalistic instances will be grammatical when removing w_2.** The w_2 bias is applied to the naturalistic instances by adding a prefix (or suffix) to the text. This prefix (or suffix) does not change the reasoning about the underlying task.
>
> For example, for the lexical inference task, a prefix of either ‘what we think’, or ‘what we believe’ is added at the beginning of the sentence (see line 353 and the footnote 4 on page 5). This prefix replaces the word ‘What’. The w2-bias correlates with the entailment class when both prefixes are the same (both sentences either have ‘what we think’ or ‘what we believe’ at the beginning). On the other hand, the w2-bias correlates with the non-entailment class when both sentences contain a different prefix.
>
> For the ‘bias-neutral’ examples (w_2=0), there is no prefix added at the beginning of the sentences.
>
> We shared below an example, using the hypothesis and premise mentioned by the reviewer. We will clarify this in the final version.
>
> | Setting     | Premise | Hypothesis |
> | ----------- | ----------- | ----------- |
> | Bias-neutral (w_2 = 0)    | What the group hate most is the sickness     | What the group hate most is the illness |
> | Bias correlates with entailment (w_2 = 1) | What we think the group hate most is the sickness | What we think the group hate most is the illness |
> | Bias correlates with non-entailment (w_2 = 2) | What we think the group hate most is the sickness | What we believe the group hate most is the illness |

---

### Official Review · Reviewer_BvPV · 2023-08-05

**Soundness:** 3

**Excitement:**

3: Ambivalent: It has merits (e.g., it reports state-of-the-art results, the idea is nice), but there are key weaknesses (e.g., it describes incremental work), and it can significantly benefit from another round of revision. However, I won't object to accepting it if my co-reviewers champion it.

**Paper Topic And Main Contributions:**

The authors present analysis on how methods for "debiasing" models based on features can lead to greater biases on untracked features. They demonstrate this on a mix on synthetic and natural language tasks, synthetically injecting the features to experiment on, and on BERT and DeBERTa models.

**Questions For The Authors:**

N/A

**Reasons To Accept:**

The experimental setup is well described and well-motivated, and the results are compelling.

**Reasons To Reject:**

One concern I have is that synthetically injected features may not share properties in practice with actually naturally occurring features/biases picked up by the model.

**Reproducibility:**

3: Could reproduce the results with some difficulty. The settings of parameters are underspecified or subjectively determined; the training/evaluation data are not widely available.

**Reviewer Confidence:**

1: Not my area, or paper was hard for me to understand. My evaluation is just an educated guess.

---

> ### Author Rebuttal · Authors · 2023-08-29
>
> Thank you for engaging with our work!
>
> **We are glad you found our results compelling and our experimental design well-motivated.**
>
> > Synthetically injected biases may not share properties with naturally occurring biases
>
> This has been addressed for the following reasons:
> 1. The data used in this work remains close to natural examples.
>
>  Bias feature types have been substantiated in the literature (word overlap and tigger word biases [Dasgupta et al., 2018; Naik 129 et al., 2018; McCoy et al., 2019.]) (L 128-129; L 348-351). Task features include features relevant to real NLP tasks (DPR commonsense reasoning [Rahman and Ng, 2012], Lexical Entailment).
>
> 2. The synthetic simulation allows control of several factors that have thus far made it challenging to conduct a similar analysis of debiasing methods, and we outline the limitations of this approach in the limitations section [L 577-586]. In NLP datasets, there may be several interacting confounds [L 61-71]. Thus, we sought to make a systematic simulation which allowed us to study the actual behavior of debiasing methods in a way that has not been done thus far.

---

### Meta-Review · Area_Chair_Nv3N · 2023-09-16

**Recommendation:** 4

**Metareview:**

This paper examines 3 debiasing techniques for neural networks. They find that the techniques indeed result in models that rely less on the biases aimed to be removed, but may cause them to instead start relying on other unknown and unidentified biases, rather than helping it robustly solve the task.

The reviewers are generally in agreement that the question as well of the results are interesting. Both BvPV and FLhX point out that the synthetic nature of the used data takes away a bit from the strength of the results. I do not find that the authors succesfully rebut this concern (and neither does BTLE). Nevertheless, I don't think this should stand in the way of acceptance. There is always a trade-off between using realistic data and synthetic/toy data in terms of the conclusions that can be drawn, and I think the paper presents a valuable contribution as is.

---

### Decision · Program_Chairs · 2023-10-07

**Decision:**

Accept-Findings

**Comment:**

This paper examines 3 debiasing techniques for neural networks. They find that the techniques indeed result in models that rely less on the biases aimed to be removed, but may cause them to instead start relying on other unknown and unidentified biases, rather than helping it robustly solve the task.

The reviewers are generally in agreement that the question as well of the results are interesting. Both BvPV and FLhX point out that the synthetic nature of the used data takes away a bit from the strength of the results. I do not find that the authors succesfully rebut this concern (and neither does BTLE). Nevertheless, I don't think this should stand in the way of acceptance. There is always a trade-off between using realistic data and synthetic/toy data in terms of the conclusions that can be drawn, and I think the paper presents a valuable contribution as is.